# Pandemic Preparedness: A Scoping Review of Best and Worst Practices from COVID-19

**DOI:** 10.3390/healthcare11182572

**Published:** 2023-09-18

**Authors:** Alessia Maccaro, Camilla Audia, Katy Stokes, Haleema Masud, Sharifah Sekalala, Leandro Pecchia, Davide Piaggio

**Affiliations:** 1School of Engineering, University of Warwick, Library Rd., Coventry CV4 7AL, UK; katy.stokes@warwick.ac.uk (K.S.); leandro.pecchia@unicampus.it (L.P.); davide.piaggio@warwick.ac.uk (D.P.); 2Global Sustainable Development, School for Cross-Faculty Studies, University of Warwick, Library Rd., Coventry CV4 7AL, UK; camilla.audia@warwick.ac.uk; 3Institute of Advanced Studies, University of Warwick, Library Rd., Coventry CV4 7AL, UK; haleema.masud.1@warwick.ac.uk; 4School of Law, University of Warwick, Library Rd., Coventry CV4 7AL, UK; sharifah.sekalala@warwick.ac.uk; 5Department of Engineering, University Campus Bio-Medico of Rome, Via Alvaro del Portillo 21, 00128 Roma, Italy

**Keywords:** preparedness, COVID-19, pandemic, best practices, scoping review, ethics, health practices, science–policy–society interactions

## Abstract

The COVID-19 pandemic highlighted the scale of global unpreparedness to deal with the fast-arising needs of global health threats. This problem was coupled with a crisis of governance and presented in the context of globally hitting climate crisis and disasters. Although such a pandemic was predictable due to the known effects of human intervention on the surrounding environment and its devastating secondary effects, such as climate change and increased zoonoses, most countries were unprepared to deal with the scale and scope of the pandemic. In this context, such as that of the climate crisis, the Global North and Global South faced several common challenges, including, first and foremost, the scarcity of resources required for health, policy, wellbeing and socioeconomic wellness. In this paper, we review the most recent evidence available in the literature related to pandemic preparedness and governance, focusing on principles and practices used during the COVID-19 pandemic, and we place it in the context of a European Parliament Interest Group meeting (this event took place on 21 March 2023 during the “European Health Tech Summit”) to ground it within ongoing discussions and narratives of policy and praxis. The review identified key practices and principles required to better face future health threats and emergencies. Beyond health practices relying on technology and innovation, it is useful to mention the importance of contextualising responses and linking them to clear goals, improving the agreement between science and policymaking, thus building trust and enabling transparent communication with the general public based on clear ethical frameworks.

## 1. Introduction

The COVID-19 pandemic highlighted gaps in both national healthcare systems and wider preparedness policies. The post-pandemic Global Health Security (GHS) index report confirmed that no country was fully prepared to tackle an emerging public health emergency threat [1]. The difference between the GHS index ranking and the real-world performance of countries, as determined via COVID-19 performance indicators, suggests that the GHS index might have underestimated the preparedness level of certain countries while overestimating that of other countries [2]. While some joint strategies were adopted at an international level, similar challenges stemming from the same emergency were tackled in different ways across the globe. It is now evident that inadequate consideration of the contextual cultural particularisms of different countries undermines the uptake of specific measures; in this paper, we reflect on the lessons learned and discuss ways in which they can support future preparedness [3].

Despite the challenges involved, it is possible to envisage the global exchange in knowledge and a universal approach (open to all) to pandemic preparedness and governance that considers different local contexts. Some authors of this paper have previously published various investigations in pursuit of this goal, promoting the application of the fields of biomedical engineering and ethics to biomedical engineering, with specific focus on frugal medical device design and regulation and pandemic management [3,4,5,6,7]. While there has been progress in sharing medical data, the authors argue that a globalised response, which would be grounded in contextualised approaches, can be achieved if there is buy-in to move beyond a global (but too generic) versus siloed (with no globalised view) divide. The response to COVID-19 demonstrated that a generic uncontextualised approach can lead countries to feel less represented in general decisions and turn to self-referentialism, causing friction in the uptake of some health measures (e.g., vaccine hesitancy in some countries) [3,7]. Furthermore, the management of the COVID-19 pandemic highlighted a lack of interconnectivity between countries and the absence of an up-to-date interoperable legislative framework to allow countries to collectively, efficiently, and rapidly respond to the pandemic [8]. Finally, subsequent analyses of metrics and indicators of the GHS further indicated that a country’s response to prior health threats should be incorporated into future GHS index reports [2]. In sum, COVID-19 highlighted the fundamental importance of reflecting on the possibility of preparing for the future (i.e., preparedness) and drawing lessons from the past. Creating spaces for reflexivity and learning has been put forward as a key element in sustainable, just, and long-term resilience in terms of disaster risk reduction [9]. In this review, we understand preparedness to be a set of precautionary measures to be taken in case of potential disasters and a key aspect of emergency management; the current situation, in the aftermath of a pandemic, offers space for learning and reflection for the future.

According to the United Nations, disaster preparedness involves “forecasting and taking precautionary measures before an imminent threat when warnings are possible” [10]. Preparedness is a relatively young concept, and its appearance is linked to a shift in the approach relative to disasters (e.g., hurricanes, pandemic, etc.), namely from a reactive approach enacted in response to calamities to a proactive approach, with measures put in place to contain possible catastrophes. Despite plenty of empirical examples and practices, preparedness is still lacking at the theoretical level, mainly due to a lack of conceptual refinement and terminological agreement [11,12,13,14,15]. Nevertheless, despite facing pandemics since they were first recorded (i.e., the Plague of Athens in 430 BC), global preparedness (prevention, prompt response, and restoration) is lacking [16]. It is noteworthy that in March 2020, Google trends reported the highest peak of interest in pandemic preparedness since 2004, with this moment being aligned with COVID-19 outbreak. There has also been increased governmental and scientific interest (e.g., the Engineering X Pandemic Preparedness programme led by the Royal Academy of Engineering) [17].

Publishing guidelines and tools for enacting emergency responses (e.g., infection prevention and control (IPC)) during the COVID-19 pandemic proved to be insufficient to warrant effective preparedness and adequate governmental response. One of the alleged reasons behind this is the much-criticised slow and cautious approach of the World Health Organization (WHO) in terms of warning about the human transmissibility of the virus, declaring a Public Health Emergency of International Concern (PHEIC), and endorsing the public use of face masks and other containment/prevention strategies. Governments did not outshine the WHO, as their inter-coordination was also inadequate to this respect [18].

However, the WHO, despite welcoming the general recommendations presented in the paper by Sachs et al. [18], does not agree with some parts of that report, which are considered to be “omissions” and “misinterpretations”.

For the purposes of this paper, we aim to look at measures and practices of countries who reacted, responded, and restored their systems during the COVID-19 pandemic. This study is rooted in the consideration that being prepared, in the sense of having an absolute, normative, or prescriptive plan to control future pandemics, is ultimately unachievable. Moreover, in our ever-more globalised world, the risk of future pandemics is higher than ever [19]. Effective governance and decision-making rely on the outcomes and responsible communication of scientific research (e.g., concerning the characteristics of the pathogenic threat or the efficacy of IPC strategies) conducted with rigour and integrity, regardless of the urgency and pressure of the situation.

This article situates the findings from the literature within the context of a European Parliament Interest Group (EPIG) event that took place on 21 March 2023 during the “European Health Tech Summit”.

This article, therefore, collects the evidence available in the literature related to preparedness and governance during the COVID-19 pandemic, focusing on lessons learned for future policies and worst and best practices. We used the proceedings of this meeting to triangulate our review and ground it in the ongoing praxis regarding learning from COVID-19 and reflecting on practices to be better prepared, as well as more resilient, in the event of a future health threat [9]. Future work may include a Delphi study to scope and raise consensus on these recommendations among international experts to co-create a theoretical–practical framework to guide the transdisciplinary management of future healthcare emergencies.

## 2. Methods

This paper presents the results of a scoping review triangulated with results of the EPIG event; the organisation of and participation in the event was supported by the Policy Support Fund of the University of Warwick.

### 2.1. Scoping Review

A scoping review was performed. In order to check whether our work would be original and not duplicating possibly already existent works, before starting this study, we performed an initial check on Scopus to analyse the existing reviews on the same topic. Since no similar work was found, we proceeded with the aforementioned scoping review.

### 2.2. Search Strategy

The scoping review was conducted according to the PRISMA guidelines for performing scoping reviews (Preferred Reporting Items for Systematic Reviews and Meta-Analyses) [20,21]. The review was performed following the Population (P), Concept (C), and Context (C) model [22]:P—Governance and preparedness practices;C—Lessons learned and best and worst practices;C—The global COVID-19 pandemic.

A systematic search was conducted on Scopus in February 2023, using the search string displayed in Table 1. The search string was combined with the AND operator for three main topics, i.e., the pandemic, policies, and best/worst practices. Scopus was chosen due to the fact that it is a database that also includes other databases (e.g., MEDLINE) and collects texts of a medical, scientific, technical, and social nature.

### 2.3. Study Eligibility

The only inclusion criterion was as follows:Scientific articles focusing on the management of the COVID-19 health emergency.

The exclusion criteria were as follows:Articles not published in the English language;Articles in which the full texts were not accessible;Articles published before 2019;Article of the following types: letters to editors, editorials, commentaries, and review articles;Articles focusing on topics other than the political management of pandemics (e.g., military lessons or specific clinical interventions);Articles reporting specific case studies (e.g., geriatric patient management);Articles reporting modelling studies.

### 2.4. Study Selection

Two authors independently screened the studies by title and abstract, while three authors completed the full text screening based on the inclusion and exclusion criteria, with conflicting decisions being mitigated by an additional independent reviewer. Relevant data were then extracted using an ad hoc extraction table to facilitate the analysis and narrative synthesis.

### 2.5. Data Extraction

Relevant data were extracted and collected in an ad hoc Excel sheet and organised by author, year, title, geographical focus (countries or global), theme (infodemic, evaluation of intervention, governance, etc.), keywords (/subtheme), aims/objectives, applied methods, practices/interventions studied, key findings, best practices, worst practices, and any other findings related to informing evidence-based policy. The summarised version of this sheet is presented in Table 2 and Table 3.

### 2.6. Data Synthesis

To synthesise the extracted data, a narrative synthesis method was used [47]. This approach enabled the authors to organise the results according to practices and principles, which were compared to and contrasted with the wider literature. This method allowed us to make further considerations, which are presented and contextualised in the discussions section below.

### 2.7. Triangulation Phase

On 21 March 2023, an EPIG was held during the “European Health Tech Summit”. This event took place in the form of two sessions (i.e., “Innovation and technology as gateways for a safer and healthier future—The impact of COVID-19 and lessons learned” and “Ensuring European and global preparedness for future crises”) with a panel of experts presenting cutting-edge research related to COVID-19 and telemedicine, followed by a dedicated Q&A slot. The event was hosted live and online to reach a wider audience. Speakers at the event include Members of the European Parliament, academics from European universities, and members of the European Commission. The sessions looked at the impacts of the COVID-19 pandemic on critical devices availabilities, discussed the importance of societal connections and broad cooperation, and debated the importance of telemedicine. The event was co-organised by the European Alliance of Medical and Biological Engineering and Science (EAMBES), the University of Warwick, and Member of the European Parliament Dr. Stelios Kympouropoulos (European People’s Party Coordinator of the Special Committee for COVID-19 (COVI) and Vice-Coordinator of the Committee for Public Health (SANT)).

The authors attended the event and used its notes and proceedings to ground the scoping review’s results in ongoing debates around science–policy–society interactions to improve preparedness. In particular, the authors focused on some of the discussions related to the potential of telemedicine and e-health solutions to revolutionise healthcare delivery (and the challenges); the ideas on pandemic management, which focused on users’ and media perspectives during a pandemic; the importance of bridging a gap between theoretical solutions (clinical guidelines) and real-world implementation; and, finally, the role played by collaboration during a pandemic. These themes were critically reflected upon, used to situate the scoping reviews’ findings, and presented in the discussions.

## 3. Results

The search on Scopus and the study selection process is illustrated in Figure 1. Indeed, the search returned 903 records, of which 24 met our inclusion criteria.

The characteristics of the included studies are illustrated in Table 2. Most studies provided a narrative analysis of practices related to preparedness or pandemic response strategies. Three studies also used statistical methods to further analyse the impacts of practices [28,30,46]. Two studies gathered perceptions towards the deployment/communication of practices from groups such as healthcare professionals [37,45]. A broad range of practices was addressed, ranging from specific containment measures (e.g., social distancing, contact tracing, etc.) to overall governance strategies (e.g., the political ideology, high-level coordination of practices, etc.). These studies presented a vast number of practices and principles, ranging from public health control measures to public attitudes, as well as from media communication to government action. This paper aims to analyse them from a preparedness point of view; in other words, we draw on biomedical engineering, bioethics, and political ecology theories to understand how these actions, tools, principles, and practices can lead to better outcomes in the future. In the Anthropocene, which is characterised by ‘wicked problems’, preparedness has become an ever-more important pillar of sustainable disaster risk management practices [48]. Addressing complex issues requires a transdisciplinary approach, in which insights from various disciplines are harmonised, drawing from diverse expertise to foster novel insights [49]. Traditional siloed disciplines may offer deep knowledge but can fail to recognise the interconnectivity of systems involved. In the context of preparedness during the COVID-19 pandemic, this transdisciplinarity approach aids in understanding the intricate intersections of ecology, urbanisation, public health, socio-economic factors, and global connectivity. This paper embraces a transdisciplinary perspective, which will enable holistic strategies that account for the multifaceted nature of the challenges faced in the Anthropocene [50].

Based on the existing literature and guided via thematic analysis, we present the results in two different sections, i.e., practices and principles. Table 3 summarises areas for improvement, specific lessons learned, and opportunities for future action derived from each paper.

### 3.1. Practices

In this section, we present data regarding COVID-19 management practices, such as science–policy communication, contextualising responses, innovation technologies, and health practices.

#### 3.1.1. Science–Policy–Society Communication

The COVID-19 pandemic brought significant public attention to the role played by science in policy-making, as well as the importance of the local socio-cultural context. Evans [42] presented lessons learned, from the perspective of the UK, from science-driven policy-making and the communication of policy decisions to the public. The study presented the challenges involved in establishing the reasonable levels of evidence required to undertake certain decisions, particularly when facing novel crises. The article noted that there is a difficult trade off in terms of the level of evidence required for a policy decision and the time taken to reach that decision and enact an effective policy. The paper also showed that it can be very difficult for governments to request the ‘right kind of advice’ from the scientific community—a challenge which may be remedied through better policy–research interactions and coordinated efforts. This study also addressed the need to improve the communication of science in general, in particular in relation to the communication of uncertainty. In a similar vein, Irwin [24] highlighted the need for the media to capture different ideas at different moments of the pandemic. This study stated that science–policy struggles did not appear in the news and the media generalised the impact of COVID-19, failing to report differences in its impact between different regions. According to Irwin, the media does not always distinguish between expertise, data, facts, and science, which is key for building trust between governments, populations, and the scientific community and reducing a pandemic’s impact on mental health.

Similar challenges in terms of widespread sensible communication were found by Upadhyay et al. [45], who analysed the perceptions of healthcare professionals in Indian Technical and Economic Cooperation (ITEC) countries towards the preparedness and responses of their countries during the COVID-19 pandemic. The top three most reported challenges were a lack of awareness among the public (67%), the undertesting of the susceptible population (81.4%), and a lack of appropriate personal protective equipment (PPE) (71.1%). Min et al. [28] explored communication in a similar way and looked at cultural perspectives and their impacts on control measures in the context of COVID-19 in Organisation for Economic Cooperation and Development (OECD) countries. Their argument was built on striking evidence that shows how a relationship between a nation’s cultural dimensions and its COVID-19 efficiency scores were important. The paper puts forward the idea that focusing on country-wide measures is not a particularly efficient approach; socio-economic context is key in finding appropriate measures, which must be tactfully communicated, to encourage a greater uptake and, ultimately, higher efficiency in pandemic management.

Bartels et al.’s [37] work on the case of North Carolina also focused on how public health officials communicated with each other and the general public. The state developed an interdisciplinary rapid-message testing model for COVID-19 to quickly create, test, and share messages with public health officials for use in health campaigns and policy briefings. The model focused on motivations for social distancing, rather than barriers to compliance, because behavioural scientists argue that how much a message motivates or discourages action is strongly correlated with actual behaviour and, therefore, provides a promising entry point for health behaviour decisions. Their study reported that survey participants rated messages focused on protecting themselves and others higher than those focused on norms and fear-based approaches. In fact, pairing behaviours with motivations increased participants’ desire to respect social distance measures across all themes and subgroups. Overall, this interdisciplinary model was a good example of rapid-message testing that reduced the time needed to deliver evidence-based messages and increased the relevance of research for policy makers and public health officials. However, the proposed model also has several limitations, such as key behaviours across the country changing as the pandemic evolved and the difficulties involved in achieving a representative sample in surveys. These characteristics limited the generalisability of the findings to the target audience.

#### 3.1.2. Institutional Fragmentation: From Local Responses to Global Outcomes

Several of the identified studies emphasised the benefits of local and flexible responses to outbreaks. Zhang et al. [46] provided a comparison between non-pharmaceutical policies enacted in China and Germany in response to COVID-19, emphasising that policy choices reflected the differing goals of the two countries. In their work, they state that China’s aim was eliminating the virus, which was reflected by the employment of more stringent policies, such as locking down the worst-hit areas and initiating residential closed management. In contrast, Germany focused on restricting gatherings and contacts to reduce transmission, as their aim was the mitigation rather than elimination of diseases, and more specifically, the protection of high risk-groups. All of the analysed policies were shown to be effective, as they were all associated with a reduction in cases at different levels. For example, in China, the expansion of medical insurance coverage to suspected patients granted the highest association with a reduction in cases, while in Germany, the highest association was found for the ‘no-contact protocol’.

Agnew [34] showed the case of the Unites States government, where the conflict between different political ideologies (i.e., nationalism, federalism, etc.) and the politicisation of the pandemic, e.g., the use of the pandemic by President Trump for electoral purposes, led to the mismanagement of the healthcare emergency. This chaotic administrative approach and the conflict between the decentralisation and centralisation of management without coordination across tiers of government, according to the author, should have been replaced with a more polyphonic practice of federalism that would have led to better management of the pandemic. In turn, Moeenian et al. [29] proposed a different take based on a specific practice implemented in Iran. In this study, the focus is on the roles played by Non-Governmental Organisations (NGOs) in the Global South or emerging economies, as well as how they can be useful in the context of pandemic preparedness. The authors found that if the policies of existing bodies were aligned with those of NGOs, there was less of a chance of duplication and more efficient management and division of tasks. The study suggests that governments should establish institutions to facilitate communication with NGOs as, in some contexts, they have more speed and agility to tackle pandemics locally and influence national levels of efficiency. Similar results in terms of the higher efficacy of high political engagement combined with layered coordination were found by Ngoy et al. [30], who looked at coordination mechanisms that were used in the early stages of managing the COVID-19 pandemic in the WHO’s AFRO region.

Along similar lines, we considered Pennestrí et al.’s [31] evaluation of Lombardy’s response to the pandemic, which strove to improve coordination of not only overall institutional and governmental structures, but also healthcare facilities. The authors proposed to do so by leveraging telemedicine technology, especially in the early stages of a pandemic, to allow the remote monitoring and treatment of non-severe patients unless direct contact was necessary. Another key argument was made in the paper regarding the private medical sector and the need for clear requirements to be respected by private providers to tackle the cherry picking of patients and funding, as these have issues negative impacts on public health provision. In a different context, but putting forward a similar argument, we found Prajitha et al.’s [32] paper on the Indian State of Kerala’s initial response to the pandemic. The authors agreed with arguments outlining a need to reduce institutional fragmentation and push the analysis to prove that the impact of synergy between social capital, robust public health systems, participation, and volunteerism lead to stronger health system preparedness. Kerala’s example was brought forward as the government, learning from responses to past viral outbreaks, was able to base its healthcare on social justice and equity, including public–private partnerships that ensured adequate manpower and material resources, combined with community participation and awareness. Braithwaite et al.’s [38] paper also provided interesting conclusions around health practices. In their cross-sectional study of 40 health systems’ responses to COVID-19 (36 countries in the OECD area, plus Singapore, Malaysia, Taiwan, and Iran), they looked at data up to April 2020 regarding each government’s capacity to respond to a pandemic, stringency measures, and approaches to testing, as well as COVID-19 cases and deaths. The authors highlighted that even in situations in which a national government’s pre-pandemic capacity to respond was lacking, successfully adopting early stringent public health measures in response to COVID-19, such as testing and tracing, still made a substantial difference. In line with the other literature, the study shows that stringent measures are not sustainable in the longer term and broad-based testing and tracing was key in managing the virus. An interesting perspective is given around a government’s capacity to plan for different socioeconomic, cultural, and ethnic backgrounds, as policies will affect various people differently. In particular, the authors highlight the negative and knock-on effects of lockdowns in terms of the economy, as well as social justice.

#### 3.1.3. Health Practices

This review would not be complete if it did not address best and worst practices regarding health measures prior to, during, and closely after the pandemic. Here, we present key data from authors who analysed these measures in different countries. Goodyear-Smith et al. [43] compared COVID-19 preparedness and responses in four countries (Australia, South Africa, Egypt, and Nigeria). A key finding of the study was the crucial role played by an integrated response between primary care and public health services in responding to the pandemic. The authors noted this finding has long been recognised as a crucial element of epidemic response. The study also found that there was inequity in the vaccination strategy, as well as testing, between High- and Low-Income Countries. This issue was demonstrated by the reduced capacity for testing in Nigeria and Egypt in contrast to Australia. Saleh et al. [33] highlighted that in order to improve pandemic responses, there must be accurate documentation of the strategies employed and lessons learned from previous outbreaks. In Nigeria, the Framework for a Public Health Emergency Operations Centre (PHEOC Framework) outlined by the WHO was used to create hubs for stakeholders across the public health structure in order to provide a platform for the learning, training, and documentation of practices. Ansah et al. [35] analysed the Singaporean Government’s intervention in the management of COVID-19 pandemic, which prioritised the mitigation strategy (which aims to limit movement at the population level; social distancing/community lockdown) to that of containment (quarantine based on contact tracing or their location). The authors stated that contact tracing, testing, and aggressive containment are key procedures that should be combined with social distancing, which is vital in slowing COVID-19, but much less effective when used alone. Among the best practices used to suppress the number of COVID-19 infections in Singapore, the authors pinpoint the following examples: (a) the timing of the intervention; (b) the contact tracing, in which Singapore had a strong experience learned from SARS and physical and operational infrastructure; (c) the revision of the Infectious Disease Act (IDA), which ensures that all measures needed to control any future outbreaks could be implemented. Along similar lines, Lee and Lim [26] put forward the idea that medical and economic measures should always come together in the case of viruses similar to COVID-19, which require containment or lockdowns to be effectively managed in the early stages of a pandemic. This approach is in line with the articles identified, which argue for context-specific measures and use Data Envelopment Analysis to show the way in which restrictions, when combined with sufficient and appropriate income support, livelihood aid, and public campaigning to inform and educate, made countries perform better from both economic and medical perspectives.

More specifically, in terms of correct practices, Atsawarungruangkit et al. [36] compared the criteria used to identify suspected cases of COVID-19 in 10 countries across Asia, Europe, and North America (China, Germany, Iran, Italy, Japan, South Korea, Taiwan, Thailand, the United Kingdom, and United States of America). Moving from the consideration that the rapid and accurate identification of suspected cases is critical in slowing spread of the virus that causes the disease, the authors aimed to highlight discrepancies in the various criteria used by international agencies and highly impacted individual countries around the world. The authors show that there was no one-size-fits-all guideline in this pandemic, and no best practice criterion has yet been defined. Every country has set its own criteria based on the principle of ALARA (As Low As Reasonably Achievable) based on available resources and the situation of the country, including budget, economic impact, insurance coverage, etc. The criteria defined by all of the reviewed countries were focused on specific symptoms and epidemiological risk assessment and may fail to capture or severely under-represent certain populations, including (a) asymptomatic cases, (b) patients with a financial barrier to accessing laboratory tests (owing to a lack of insurance coverage and high testing costs), and (c) patients with a legal barrier to accessing the health care system (including undocumented immigrants and homeless individuals). The proportion of cases in the latter two groups is dictated to a large extent by the government policies of individual countries. This paper clearly highlighted the need for the coordination of efforts not only by public health organisations, but also the public and private sectors, including health care systems and the insurance industry, and most importantly, citizens themselves.

#### 3.1.4. Innovation Technology

Only two studies directly addressed the roles played by healthcare innovations in response to pandemics.

Halfmann et al. [44] proposed a theoretical framework for the creation and management of innovations in healthcare and Information And Communication Technology (ICT). The authors suggest 11 steps, which are outlined in an “innovation wheel”, focusing on monitoring, analysis, and development, as well as innovation management. Guidance is provided for each task to improve the innovation process and strengthen the systematic early dialog between stakeholders, especially between the Global North and Global South, which was found to be key in the process. This paper offers a framework to build capacity for innovation dimensions (such as partnership mobilisation, evaluation and monitoring, literacy, etc.) and emphasizes the active engagement of all stakeholders. This method is an interesting and novel instrument to help overcome current and future barriers in planetary health innovation management and support potential breakthrough discoveries in ICT. Goodyear-Smith et al. [43], tangentially to their main focus, also found that among all four countries evaluated, there was rapid adoption of telehealth in response to COVID-19. Telehealth was used to facilitate contact tracing and reduce the number of transmissions in health facilities. However, it is noteworthy that the ability to leverage technology and innovation is context dependent; Coral-Almeida et al.’s [41] study showed how the pandemic has negatively impacted digital access through an analysis of the management and impact of COVID-19 in Ecuador. The authors noted a widening of the digital divide and a need for a policy platform that promotes digital literacy and access, particularly among less advantaged socioeconomic groups.

### 3.2. Principles

In this section, we present the results of studies that address the risks of framing policies/interventions in terms of so-called ‘best practices’ [39,40]. For example, initiatives such as prescribed social distancing or isolation are entirely unfeasible if applied in crowded living situations. Instead, approaches should be localised, co-produced, and bottom-up in nature to ensure that effective practices are upheld without unwanted economic and social consequences.

#### 3.2.1. Building Trust and Ways of Communication with the General Public

Canario Guzman et al. [39] called for a strengthening of research ethics and regulatory frameworks to facilitate strategic policy decisions that coordinate research efforts, aligning with priorities and ensuring accountability and transparency. The authors highlighted the importance of collaboration and knowledge sharing, both within and between national regulatory bodies. These findings were consistent with those of Chowdhury and Jomo [40], who emphasised that transparency and coordination in policymaking are crucial for building and maintaining trust between citizens and government. Mersha et al. [27] showed an interesting and specific aspect of trust-building activities, i.e., between government and healthcare professionals. Their study showed that in the context of South Omo (Ethiopia), there was a gap between the attitudes towards precautionary measures and their implementation in practice. In their paper, they argued that capacity-building activities aimed at healthcare professionals are a core part of pandemic preparedness and should be provided to ensure that the general public can follow them by copying their attitudes and actions.

#### 3.2.2. Ethical Guidelines to Mediate the Relationship between Science and Policy-Making

Only two papers specifically addressed ethical concerns. Herstein et al. [23] described the functions of an existing preparedness network for global infectious diseases, focusing on the importance of rapid information exchange, which allowed the rapid adoption of treatments and protocols. The authors argued that using pre-existing or repurposing older networks as platforms for sharing preliminary information and giving access to real-time data before they are available in scientific or medical communities is a crucial step to take when preparing for further pandemics. Jegede et al. [25] argued that a framework and ethical guidelines are extremely valuable during a pandemic; building on the discourse around contextually sensitive measures, the authors showed that ethically sensitive communication and appropriate countermeasures had a positive impact on the public in the Global South.

## 4. Discussion

Preparedness strategies for health-related emergencies prior to COVID-19 were largely overlooked, leaving communities underequipped [16]. The literature and scoping reviews have shown that the time during a pandemic is not an ideal situation for building and training preparedness in terms of either resources or ethics [51]. However, as COVID-19 is no longer a PHEIC, looking back in hindsight and analysing practices and summarising the lessons learned from this major health challenge is now essential to improve our preparedness and foster evidence-based policymaking [9,52]. After analysing the existing literature reviews published since 2019 on the topic of interest (following the same search strategy presented above), it was found that only five were systematic. These reviews focused on the effectiveness of different strategies in terms of preventing the spread of COVID-19, and most of them included data up to 2020. In this context, the EPIG event helped to frame the literature and discussion by grounding it in existing and ongoing debates on preparedness practices, concepts, and principles. We are now able to learn more about how the world reacted to COVID-19 and which best and worst practices are emerging from the management of the pandemic worldwide.

Science–policy–society interactions play crucial roles in shaping the landscape of global health principles and practices. This review argues that post-pandemic reflections can help in bridging the gap between scientific research and public policy, especially if guided by community-led, culturally sensitive, and context-specific approaches.

Intuitively, we understand that state-of-the art research and the best available data and evidence should be used to guide public health decisions; however, in practice, the science–policy–society interface is much more complex. This review shows that an embedded model of communication, where there is specific attention to the roles played by scientific accuracy, policy-making needs, and societal contexts [53] as connected and communicating processes, is key in managing a pandemic. The EPIG event reported on the importance of interdisciplinary and sustainable collaboration amongst stakeholders, including policy makers, biomedical engineers, scientists, and society. This collaboration is key in terms of achieving sustainable and equitable data access and sharing, as well as advancing ideas for managing pandemics that can be easily adapted to diverse local contexts, moving beyond the Global South/Global North divide. In line with this European Parliament-level debate, we argue that in order to achieve harmonious communication between science, policy, and society, we need transdisciplinarity [54], as well as context-specific solutions, to improve communication. In attempting to avoid a siloed debate around a specific topic, we aimed to develop a Concept of Global Health that was in line with the latest declarations related to Agenda 2030 and the Sustainable Development Goals [55]. This method also offers interesting overlaps with the latest responsible co-production of knowledge approaches, acknowledging that communities (and, generally, society) should be at the forefront of knowledge co-creation to ensure their culture, priorities, and behaviours are respected, which, in turn, ensures a useful, usable, and, ultimately, used preparedness response [9,56,57].

There were extreme differences in pandemic management not only between countries, but also between administrative areas (states, regions, and municipalities) within countries. The review puts forward two key examples (Italy and the USA), where a regional fragmentation and a federal/state dichotomy prevented the more effective implementation of pandemic measures. In line with current research, we argue that these times of crises show systemic weaknesses that can be addressed once the emergency is called off, meaning that the countries can be better prepared for the next potential crisis [9,58]. Moreover, the politicisation of the decision-making process related to COVID-19 and the consequent impact on death rates and economic measures opened up significant questions about international law and cooperation during a pandemic [59]. The debate was also brought forward by the EPIG session, which called for a strong yet flexible global regulatory framework able to reflect rapid advancements in available technologies.

This approach is paramount in the context of the innovative technologies found in the literature. In fact, it can be noted that common themes emerge, such as the importance of contextualised approaches, which was a key finding in the included studies. We should indeed reflect on a minimum common denominator, such as Nussbaum’s “capabilities” of individuals and their “functionings” to be guaranteed and implemented in a manner appropriate to the specific context, in order to offer tailored responses to health threats [60]. Therefore, self-determination is one of the key factors: low-resource settings should shape their responses with regard to their own traditional beliefs [61], avoiding exacerbations of pre-existing gaps between the rich and the poor [62] and aiming to find a commonly shared perspective, i.e., that of human rights. The non-contextualised responses and practices can lead to no benefit and even be detrimental. For example, not taking into account the local Beninese culture of relying on traditional medicine and religious/mystical aspects or underestimating the lingering traces of colonialism slowed the uptake of allegedly “Western” approaches for COVID-19 management (e.g., plot theories of “whites” conspiring against the local population) [3]. The EPIG spoke of “responsible technologies”, which is a concept that incorporates sustainability, frugality, and social justice, meaning that healthcare innovation (digital technologies, such as contact tracing apps, telemedicine, etc.) can be globally deployed to enable accessible, affordable, and resilient healthcare that is integrated in a sustainable and equitable manner.

The included studies highlighted that there are common underlying principles that can be applied in a contextual framework to ensure fast and effective strategies and communication mechanisms. These mechanisms should be based on timely scientific evidence and available both to health practitioners and the general public. Improving communication, in this sense, also entails looking at mechanisms that maintain academic/scientific integrity while allowing the quicker turnaround of science–policy–society interactions. This approach means making full use of context-specific entry points that will promote a specific behaviour or behavioural change, using pre-existing networks and playing to countries’ strengths, which may mean many different (but coordinated) approaches, rather than a single, and sometimes inappropriate, pandemic strategy. While this concept is not new, it needs to be reiterated to ensure that further medical emergencies can be more efficiently managed.

This review also stresses the importance of fostering the creation and management of innovations in health emergencies. Indeed, this approach is aligned with a major challenge highlighted by Pecchia at the EPIG event, who sustained the inadequacy of the PPE standards and identified them as the culprit for slowing down the scaling up of the PPE production in the first wave of COVID-19. As reflected in both the literature and the discussions at the EPIG, telemedicine offers an opportunity to include the most remote and low-resource areas and reduce pressures on health services. However, there is a risk that inappropriate up-scaling of telemedicine may unintentionally exacerbate the digital divide [19]. Collaboration between countries is essential in this respect.

Finally, building trust remains a key step involved in improving the uptake of policies and measures and the willingness to adhere to regulations, as backed up by the decades of literature on social sciences [63]. While managing a crisis, decision makers and official bodies may lack the time and space to involve local communities, households, citizens, and people who will be most affected by their measures, resulting in actions that may lack ownership and seem more top down than co-produced [64]. In line with other recent studies [8], this review argues that sound ethical guidelines, ideally based on lessons learned in times of a pandemic, co-produced with relevant actors and globally generalisable (to be contextually adaptable) are a much-needed tool. This outcome stems from the need to frame science–policy–society communication in a way that keeps it grounded in data but flexible to potential bottom-up changes, easily adaptable, and ultimately useful, as well as used by the affected population. This review contributes to filling a gap and constitutes a starting point for a global reflection on the principles, ethics, and tools required to improve future preparedness.

Table 4 summarises the best and worst practices derived from the scoping review, as discussed in this study.

A few limitations of this study can be highlighted in this study. While Scopus is one of the largest citations databases covering peer-reviewed journals, being interdisciplinary in content and international in coverage, it does not include all existing evidence. Since our aim was to give an overall image of the existing practices involved in pandemic preparedness, we decided to limit our search to only refereed papers published in recognised international journals or selected conference proceedings. The results of this study could be extended by considering other indexes or grey literature. We also acknowledge a few gaps in the existing literature regarding pandemic management, including key data regarding policy collaboration and data sharing. Moreover, pandemics disproportionately affect more vulnerable populations; more research is needed not only on how COVID-19 may have impacted inequalities, but also on how preparedness can include mechanisms to avoid further unfairness. While decision-making frameworks used in pandemic planning and management are growing in use and sophistication, most policies and pathways are fragmented, siloed, and slow to integrate the needs of populations, especially marginalised groups, as they often sit outside of formal processes and structures. As we argue for a comprehensive ethical framework to be developed, we imply that research needs to boldly take the next step towards more collaborative, transdisciplinary, and transformative approaches to pandemic management, providing concrete entry points to ensure that these processes reduce, rather than reinforce, inequalities. A potential Delphi study to co-produce ethical guidelines and a practical toolbox for the transdisciplinary management of future healthcare emergencies would be a next step, as it would include participants from across global settings, disciplines, and fields.

## 5. Conclusions and Ethical Reflections

This work aimed to clarify, through a multi-methodological study of the global response to COVID-19, the best approaches to adopt during a pandemic emergency at an interdisciplinary level. Our scoping literature review, which was contextualised by the EPIG event, pinpointed key elements of best practices for pandemic management and governance (e.g., evidence based and effective IPC strategies, science–policy communication, contextualised responses, innovative technologies, and ethical guidelines). As we analysed these elements by drawing on multiple disciplines, as well as including a non-academic event, we argued that they can be considered to be a framework that could be prepared to enable the management of future health emergencies and should be placed at the core of future global conversations.

While our findings offer an essential roadmap for future health emergencies, the unpredictable nature of such events underscores a vital point: how do we practically prepare for the unknown? Sheila Jasanoff argues that the issue is related to “overestimating the certainty of our predictions and our capacity for control” [65]. Regarding COVID-19, she argues that it was the most anticipated of potential unexpected events. Moreover, she claims that “the shock of our era should remind us that such Promethean dreams [the dream to be able to master everything and outtake Nature] need to be curbed by the limits of prediction” [65].

As Jasanoff posits, perhaps it is time to shift from a purely technocratic mindset to one of “technologies of humility”. From a tangible standpoint, this approach means fostering more inclusive and diverse decision-making platforms, integrating both citizens’ voices and transdisciplinary expertise, thereby expanding the scope of perspectives within our governance structures [65].

A systematic mindset that deals with such unpredictable dangers should be cultivated using humility as a model [65]. Humility “occupies the nebulous zone between preparedness and precaution by asking a moral question: not what we can achieve with what we have, but how we should act given that we cannot know the full consequences of our actions” [65]. Humility anticipates consequences but, rather than absolving our responsibility for unforeseen consequences, “it demands that we ask in advance what new vulnerabilities might be produced by our bravest acts of preparedness, in theaters of public health, economy, environment, or war” [65]. We suggest that this approach could be combined with time and space to enable reflection and learning [9]. This multifaceted approach offers an opportunity to review and critically analyse best and worst practices with the aim of indicating a path forward. In this review, we do not predict or presume to control the unforeseen future; rather, we aim to maintain our “ethical vigilance” in order to move beyond a passive/reactive approach and towards an active and conscious disposition towards the unknown, while building sustainable, just, and equitable long-term resilience.

In reflecting on the global response to COVID-19, it is evident that a holistic and integrated approach is a necessity. The pitfalls of a siloed strategy, as observed in regional fragmentation and politicisation, underscore the imperative of seamless collaboration across disciplines, borders, and societal sectors. The merits of best practices, particularly those emphasising the harmonious confluence of science, policy, and society, underscore the importance of transdisciplinary and context-specific solutions, as well as the ethically sound co-production of knowledge. As we analyse the management of COVID-19, the overarching lesson is that true preparedness demands not only foresight, but also a unified, adaptable, and inclusive approach. Only through such integration can we hope to navigate the complexities of future health emergencies, fostering resilience and safeguarding global well-being.

In practical terms, we conclude that governance structures could benefit from creating dedicated spaces in which to perform reflective dialogues, i.e., sessions in which societal partners from diverse backgrounds critically evaluate both the successes and failures of previous strategies. These forums could serve as “learning labs/reflection spaces”, enabling us to adapt and innovate while acknowledging the inherent limitations of our foresight. Such an approach, being grounded in humility and active learning, positions us not as mere reactors to unforeseen events but as proactive stewards charting a course to enable sustainable and equitable resilience in an unpredictable world.

## Figures and Tables

**Figure 1 healthcare-11-02572-f001:**
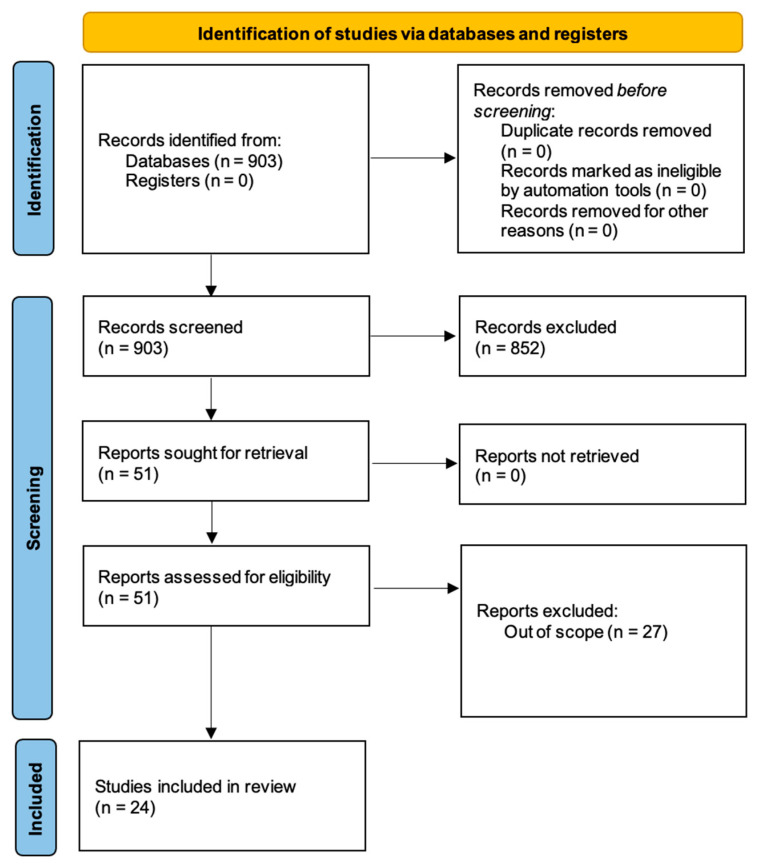
The PRISMA flowchart highlighting the selection process.

**Table 1 healthcare-11-02572-t001:** The search string used to perform the systematic search, divided into the three main topics.

Core Topic	Search String
Pandemic	(TITLE-ABS-KEY ((pandemic OR epidemic OR emergenc* OR disaster) AND (COVID* OR “SARS CoV 2” OR coronavirus OR SARS-CoV-2))
Policies	TITLE-ABS-KEY (((preparedness OR governance OR management OR prevention OR control) AND (polic* OR polit* OR guidelin* OR regulat* OR law OR decree) OR (governmental AND response) OR ((containment OR prevention) AND strateg*)))
Best worst practices	TITLE-ABS-KEY ((best OR worst) AND practice*))

**Table 2 healthcare-11-02572-t002:** Characteristics of included studies.

	Location	Study Type *	Practices Addressed
Herstein et al., 2022 [23]	Global	Narrative analysis of practices	Using a pre-existing network as a platform for managing future emergencies
Irwin, 2020 [24]	Sweden	Case study	COVID-19 responses and media representations
Jegede et al., 2020 [25]	Nigeria	Narrative analysis of practices	Measures: COVID-19 and previous epidemics/pandemics
Lee, Lim, 2021 [26]	ASEAN	Data envelopment analysis	Efficiency of IPC measures
Mersha et al., 2021 [27]	Ethiopia	Cross-sectional study	Precautionary measures conducted by health professionals (hand sanitizing, etc.)
Min, Lee, 2022 [28]	OECD countries	Data envelopment analysis	Relationship between a nation’s cultural dimensions and its COVID-19 control measures’ efficiency scores.
Moeenian et al., 2022 [29]	Iran	Grounded theory strategy	Social innovations
Ngoy et al., 2022 [30]	AFRO WHO region	Retrospective policy tracing and descriptive statistical analysis	COVID-19 response strategies, plans, regulations, press releases, government websites, and grey and peer-reviewed literature
Pennestrì et al., 2021 [31]	Lombardy (Italy)	Narrative data analysis	Regional responses
Prajitha et al., 2021 [32]	Kerala (India)	Quantitative descriptive study	Regional responses
Saleh et al., 2022 [33]	Nigeria	Narrative analysis of practices	NCDC learning mechanisms derived from the Lassa and COVID-19 outbreaks
Agnew, 2021 [34]	USA	Narrative analysis of practices	Political ideology
Ansah et al., 2021 [35]	Singapore	Narrative analysis of practices	Mitigation vs. containment
Atsawarungruangkit et al., 2020 [36]	Global; Asia, Europe, North America	Narrative analysis of practices	Case identification
Bartels et al., 2021 [37]	North Carolina	Qualitative	Message testing, rapid design, COVID-19, social distancing, emergency preparedness, etc.
Braithwaite et al., 2021 [38]	Global; 36 OECD, Singapore, Malaysia, Taiwan and Iran	Cross-sectional study	Governance approaches
Canario Guzmánet al., 2022 [39]	Central America, Dominican Republic	Qualitative	Governance approaches, ethics, etc.
Chowdhury, Jomo, 2020 [40]	Asia, South America	Case study	Containment measures (physical distancing, contact tracing, etc.)
Coral et al., 2022 [41]	Ecuador	Narrative analysis of practices	Governance practices
Evans, 2022 [42]	UK	Narrative analysis of practices	Use of evidence in policy decisions
Goodyear-Smith et al., 2022 [43]	Australia, South Africa, Egypt, Nigeria	Narrative analysis of practices	Primary healthcare policies
Halfmann et al., 2022 [44]	Europe, Africa	Narrative analysis of practices, including SWOT (Strengths, Weaknesses, Opportunities, and Threats,) analysis, surveys, interviews, etc.	Innovation governance
Upadhyay et al., 2022 [45]	13 ITEC countries	Qualitative analysis, including workshops, Delphi survey, etc.	Various pandemic preparedness strategies and responses
Zhang et al., 2021 [46]	China, Germany	Narrative and statistical analysis of practices	Non-pharmaceutical interventions

* Study types were either directly taken from the text or inferred based on the applied methods. ASEAN: Association of Southeast Asian Nations, OECD: Organisation for Economic Co-operation and Development, ITEC: Indian Technical and Economic Cooperation, AFRO: World Health Organization African Region, SWOT: Strengths, Weaknesses, Opportunities, and Threats, NHS: National Health Service.

**Table 3 healthcare-11-02572-t003:** Outcomes of the included studies in terms of areas for improvement and specific lessons learned and opportunities for future actions. NGO: Non-Government Organisation, PHEOC framework: the Framework for a Public Health Emergency Operations Centre, HIC: High-Income Country, LIC: Low-Income Country.

	Areas for Improvement	Lessons Learned and Opportunities for Future Actions
Herstein et al., 2022 [23]	Adapting to surges in capacity	Rapid information exchange facilitated the adoption of treatments/protocols. Networks can act as platforms to share preliminary findings and real-time data before their publication in scientific/medical communities.
Irwin, 2020 [24]	Faster generation of data related to disease to inform policies of disease burden	Trust between the public and the government can reduce the impact of disease control measures on mental health and incidences of abuse/domestic issues.
Jegede et al., 2020 [25]	Increasing the resilience of the health infrastructure to ensure that it is able to respond under pandemic conditions	Pandemic countermeasures need to be contextually sensitive. Communications during a pandemic need to be ethically sensitive.
Lee, Lim, 2021 [26]	Combining medical and economic measures	Restrictions should be accompanied by sufficient financial support and public campaigning to inform and educate. The cancellation of public events should be prioritised ahead of the closure of schools and workplaces.
Mersha et al., 2021 [27]	Attitudes to precautionary measures: availability and knowledge of these measures	Pandemic preparedness can be improved through capacity-building measures, training, motivation, and the recognition of health professionals.
Min, Lee, 2022 [28]	Integrating cultural considerations to country-wide measures, such as face-covering mandates and lockdowns	Outbreak responses should be tailored to the cultural traits of the country/region.
Moeenian et al., 2022 [29]	-	Social innovation can be effectively used at the national level with appropriate speed and agility to tackle pandemics. A redistributive framework and surveillance systems should be used as the basis of development, rather than to focus on economic growth. Institutions could be established by government to facilitate communication with NGOs to align the policies of executive bodies with the goals of NGOs.
Ngoy et al., 2022 [30]	-	Layered coordination is efficient at facilitating efforts, and high political engagement is key.
Pennestrì et al., 2021 [31]	Improving communication between central government directives and local healthcare systems to avoid delays in crisis management	Requirements for funding should be clearly set and be respected by private providers to avoid disproportionate investment negatively impacting public health provision. Keeping patients out of hospitals is key.
Prajitha et al., 2021 [32]	-	Healthcare-based social justice and equity should underpin outbreak responses, leveraging lessons from past viral outbreaks. Public–private partnerships ensured adequate manpower and material resources. Community participation and social capital are key to achieving successful responses.
Saleh et al., 2022 [33]	Improving the documentation of practices to make them easily available in future outbreaks	Instituting the PHEOC framework significantly improved responses. Prior successes and challenges need to be used to adapt public health responses.
Agnew, 2021 [34]	Improving coordination across tiers of government to enable a coherent response	Polyphonic federalism can mitigate the disastrous effects of a pandemic, as opposed to the either/or opposition between decentralisation and centralisation that has latterly tended to prevail in the United States. The US Army Corps of Engineers and the vaccine development programme, known as Operation Warp Speed, provided the basis for achieving excellent responses.
Ansah et al., 2021 [35]	-	Timing (rapid response) and using lessons learned from previous outbreaks is essential. Early public health measures for ‘aggressive containment’ (contact tracing and quarantine) were likely responsible for the suppression of COVID-19 cases in Singapore.
Atsawarungruangkit et al., 2020 [36]	Improving the capture of the wider population, including asymptomatic cases, patients unable to access testing, patients with barriers to health system access, etc.	Different countries demonstrate a wide variety of successful approaches. Response efforts need to be coordinated by health organisations and the public and private sectors, including the insurance industry and citizens.
Bartels et al., 2021 [37]	Improving collaboration between policy makers and researchers to generate evidence most useful for decision-making	Lean and agile principles can be applied to an interdisciplinary communication model to reduce the time required to make evidence-based decisions, linking policy makers, public health officials, and researchers.
Braithwaite et al., 2021 [38]	Improving understanding of the effects of lockdowns	The current measures and capacity of nations is insufficient to deal with pandemics. Wide testing was key to managing COVID-19. Trust in governments is key to ensure the uptake of measures. Governments must be mindful of the impact of policies (costs, disruptions, etc.).
Canario Guzmánet al., 2022 [39]	Improving health regulation and international cooperation	A research ethics system implies an ability to formulate strategic policy direction, ensure good regulation, set and monitor ethical standards, and ensure account ability and transparency. Policy should promote collaboration and joint strategies.
Chowdhury, Jomo, 2020 [40]	Adapting measures to country-specific cultural and socioeconomic contexts is needed	Recommend moving away from so-called ‘best practices’ to ensure context aware outbreak responses. Inclusive and transparent policy-making is key. Measures chosen must not perpetuate inequalities.
Coral et al., 2022 [41]	Adapting measures to context instead of using ‘cut and paste’ approaches (HICs to LICs)	Trans-disciplinarity, co-production, and localised measures are needed. The so-called ‘best practices approach’ should be avoided.
Evans, 2022 [42]	Reducing the time taken to put development measures into practice	Move towards a middle ground where usable science is understood in a holistic/sociologic manner to a ensure scientific state combined with a reasonable timeline for policy-making.
Goodyear-Smith et al., 2022 [43]	Improving equity in vaccination strategy and the availability/capacity of testing	Integrated response between primary care and public health, as well as policy-making and public health, was key to pandemic response. Telehealth was a key element of outbreak response in many countries.
Halfmann et al., 2022 [44]	-	-
Upadhyay et al., 2022 [45]	Incorporating multisectoral responses, communication and community engagement, and testing capacity are crucial	-
Zhang et al., 2021 [46]	-	Successful approaches vary depending on the overall goal of intervention and the local context.

**Table 4 healthcare-11-02572-t004:** Best and worst practices of pandemic management based on our scoping review.

Best Practices:	Worst Practices:
**Post-pandemic reflections and learning for resilience:** using hindsight to analyse practices and summarise lessons from the pandemic can improve future preparedness and evidence-based policymaking.	**Unpreparedness**: The significant oversight of health-related emergency preparedness strategies led to communities being ill-equipped during the onset of COVID-19.
**Science–policy–society interface**: A model where scientific accuracy, policy-making needs, and societal context are interconnected. This approach enhances the management of pandemics by ensuring that all stakeholders are aligned.	**Regional fragmentation**: Examples from Italy and the USA showed that regional differences in response strategies hindered effective pandemic management.
**Trust building**: it is crucial to involve communities, households, and citizens in decision-making to foster ownership and adherence.	**Politicisation**: The politicisation of pandemic decisions impacted public health outcomes and the economy, highlighting the need for international cooperation and a unified approach.
**Context-specific solutions**: approaches tailored to local cultures, beliefs, and contexts lead to more effective and accepted health responses, thus avoiding one-size-fits-all strategies.	**Lack of contextualisation**: For instance, neglecting the cultural aspects of regions like Benin slowed down the adoption of certain health practices.
**Technological innovations**: Technologies like contact tracing apps and telemedicine can be instrumental in pandemic responses. However, they must be deployed with sustainability, frugality, and social justice in mind.	**Inappropriate implementation of Technology**: there is a risk with technologies like telemedicine; if not appropriately scaled, they might widen the digital divide.
**Timely communication**: fast, accurate, and evidence-based communication mechanisms tailored to different stakeholders are pivotal.	**Infodemic**: examples show that the rapid and widespread proliferation of both accurate and inaccurate information during a pandemic hinders the ability of a government to make informed decisions.
**Interdisciplinary and transdisciplinary collaboration**: promoting sustainable collaboration between policy makers, biomedical engineers, scientists, and society is essential for equitable data access and sharing.	**Siloed approaches to pandemic preparedness and management:** when different departments, agencies, disciplines, and stakeholders operated in isolation without effective communication during COVID-19, there was a lack comprehensive and cohesive responses to the emergency and fewer opportunities for synergistic solutions derived from transdisciplinary cooperation.

## Data Availability

The data presented in this study are available upon reasonable request from the authors of this manuscript.

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
