# Peer review of "Pandemic Preparedness: A Scoping Review of Best and Worst Practices from COVID-19"

_healthcare, 2023, doi:10.3390/healthcare11182572_

Round 1

Reviewer 1 Report

The aim and nature of this study are very important. At post-pandemic era, we need to review what we have done and what we might have missed. However, the current study being a scoping review seems defocused. There are so many different types of practices regarding the pandemic control. Why have the authors included some but neglected others? The title preparedness is not accurate. Many studies reviewed simply reported the process. Controlling the pandemic involves various governmental, medical, societal and individual efforts. We cannot deal with (review) everything. Currently, the media communication, the public health control measure, the government (central/local) collaborations and some principles are mingled together randomly (simply depending on which study was described). But actually they are different things, belonging to different academic disciplines, and may have different internal requirement to make the study. It is not necessary to combine them together. This is really not a scoping review. Then the study selection criteria are also questionable. Several studies I am familiar with which I think should fall to the category of the review haven't been included (e.g. some studies published in information science field).  I would suggest that the better strategy for this study is to clearly claim certain aspect of the control measures and review the practices across countries in this aspect. 

Author Response

Thank you for your valuable feedback. 

To address some of your concerns, we’ve specified and narrowed down our specific concept of preparedness (p.2 l.90-97). 

We’ve also added a paragraph to explain, and make explicit, the remit of the study, and answer your query around multiple practices being reviewed together, and the transdisciplinarity of the authors and the paper (pp.5-6, l. 184-199). 

The methodology has been rephrased to best show our criteria for including or excluding studies (pp.3-4) 

Reviewer 2 Report

  • The aim of this research project was to analyse the COVID-19 pandemic emergency response, to identify the best and the worst worldwide approaches regarding the management and governance (evidence based and effective IPC strategies, science-policy communication, contextualised responses, innovative technologies, and ethical guidelines) and the lessons learned, in order to formulate a set of recommendations, meant to improve the preparedness for the future challenges and uncertainty management. 

  • Considering the high risk of health emergencies, not only due to pandemics but also to climate change and natural disasters, economic crisis, armed conflicts, migration, etc., such an analysis - based on a multi-methodological interdisciplinary approach, is very welcome.  

  • The conclusions regarding the humility model as potential solution for managing uncertainty and anticipating the issues, without absolving the responsibility for unforeseen consequences, and  the new vulnerabilities generated by acts of preparedness are, in my opinion, very interesting but a little bit too philosophical. I would recommend you to add some arguments/explanations regarding your own considerations  - what "an active and conscious disposition towards the unknown, while building sustainable, just and equitable long-term resilience" really means from the practical point of view?

Author Response

Thank you for your valuable feedback. 

A paragraph has been added (p.17, l. 571 to 595), to make conclusions more practical, and link them more to the findings of the paper. We provide examples of what we mean and specify some tangible potential perspectives. 

Reviewer 3 Report

Overall, the paper is well written. The work aims to clarify the best approaches during pandemic. Here, I would have some comments. From my point of view, there are not comprehensive enough exposed ideas like:

-In the event of a re-emergence of a pandemic episode, are we sufficiently prepared? 

-Can we establish the advantages and disadvantages of each method separately?

-Which one do we find more opportune, efficient, with better results?

The paper only reviews the main strategies adopted by healthcare systems. it does not provide a systematic analysis or conclusion. Ultimately, it does not  contribute scientifically to preventive behaviour or a strategy for combating a new pandemic wave.

Like the authors mentioned, the paper is not complete without the best and worse practices. Well, this are nor very comprehensive exposed. You should resume it better. The ultimate goal is to establish some good practices for the future.

Author Response

Thank you for your valuable feedback. 

In response to your comments: 

1. We have addressed this comment in the Conclusion paragraph. We’ve expanded the conclusion to comment on the newly added table (see point below) on best/worst practices and to focus on practical implications of preparedness (or lack thereof). 

2. Rather than focussing on the advantages and disadvantages separately, we’ve summarised best/worst practices into a table and addressed them from our transdisciplinary perspective; what does preparedness for a future pandemic mean, in the current and globalised world? We’ve made our reflections explicit in the discussions and conclusions – we aim to go beyond the idea of the most efficient solution to a more context-specific response with some common principles, and practices, that are in line with transformative adaptation pathways. 

3. We have added a table of generalisable best/worst practices (in Discussions, pp. 17-18). We have also amended the conclusions pp. 19-20. The table (and the paper, with this added and modified content in response to these comments) now focuses on good practices, the discussions elaborate on best principles and guiding frameworks for conceptualising, addressing and practically implementing “preparedness” in the event of future pandemics. 

Reviewer 4 Report

Article Review for “Pandemic Preparedness. A Scoping Review of Best and Worst 2 Practices from COVID-19”  

 Page 1, Abstract:  Abstract is well written and conveys to the Reader the relevance and importance of this topic.

 Pages 2, Lines 63-64:  Recommend the insertion of the word ‘and’ in the following manner to add clarity to the sentence: “In sum, COVID-19 highlighted the fundamental importance of reflecting on the possibility of preparing for the future (i.e., preparedness), and drawing lessons from the past.

 Page 2, Line 70: Recommend the insertion of a comma in the following manner to enhance better understanding by the Reader: “According to the United Nations,  disaster preparedness involves . . .”

 Page 2, Lines 86-88: It is important to first, fully spell-out any acronyms that are used in the manuscript. Recommend the Authors include the fully spell-out words for the acronyms used in the following manner: “One of the alleged reasons behind this is the criticised slow and cautious approach of the (World Health Organization (WHO) in terms of warning about the human transmissibility of the virus, of declaring a Public Health Emergency of International Concern (PHEIC), . . .”

 Page 3, Line 100: Recommend the replacement of the words “lesson learned” with “lessons-learned” for clarity as there were more than one lesson learned from the pandemic.

 Page 3, Line 128: Recommend making the word “letter” plural for accuracy. The new sentence fragment should read as follows: “. . . were letters to . . .”

 Page 4, Line 137: Recommend correcting the word “was” to “were” as the word “data” is plural.  The new sentence fragment should read as follows: “ . . . data were then extracted. . . ” 

 Page 4, Line 140: Recommend correcting the word “was” to “were” as the word “data” is plural.  The new sentence fragment should read as follows: “ . . . Relevant data were extracted. . . ” 

 Pages 6-7, Table 1. Characteristics of included studies. Table is well-organized, however some of the columns are not well-delineated from each other (e.g. some words from column one bleed over to column two, making it difficult for the Reader to comprehend what is contained in each column). Recommend the Authors ensure that words from each column do not cross over into other columns.

 Pages 7-9, Table 2. Table is well-organized, however some of the columns are not well-delineated from each other (e.g. some words from column one bleed over to column two, making it difficult for the Reader to comprehend what is contained in each column). Recommend the Authors ensure that words from each column do not cross over into other columns.

 Page 12, Line 330:  Recommend the insertion of the word “contact” to better describe what the Authors are trying to communicate. The new sentence fragment should read as follows: “ . . . (quarantine based on contact tracing or their location).”

Page 12, Line 334:  Recommend the correction of the word “pinpoints” to “pinpoint” for accuracy. The new sentence fragment should read as follows: “ . . . the authors pinpoint:”

 Concluding comments: This Reviewer commends the Authors for addressing an important topic: a comprehensive review of pandemic preparedness examining both the best and worst practices/ lessons-learned from the recent, worldwide COVID-19 pandemic. This review was comprehensive, well-organized and informative. This Reviewer commends the Authors and looks forward to this manuscript as well as future manuscript among the published literature.

Author Response

Thank you for your valuable suggestions and feedback. All your recommendations were added to the text 

Reviewer 5 Report

This review presents a comprehensive overview of the methods and findings of a scoping review conducted on the topic of COVID-19 pandemic management and governance. The authors detail their search strategy, study eligibility criteria, data extraction process, and synthesis approach. They also highlight the integration of the results with insights from an associated event (EPIG) as an attempt to enhance the relevance and applicability of the review's findings.

However, there are a few areas of critique that can be addressed:

Lack of Clarity in Methodology: While the paper mentions the use of PRISMA guidelines for scoping reviews, some aspects of the methodology could be further elaborated. For instance, more clarity on how the initial search on Scopus informed the scoping review process could provide context for readers.

Search Strategy and Sources: The search string used for the systematic search on Scopus is mentioned to be available in Supplementary Material 1, but it might have been helpful to provide it directly in the main text. Additionally, the decision to use Scopus as the sole source for the systematic search could raise questions about the comprehensiveness of the literature review.

Study Eligibility Criteria: The inclusion and exclusion criteria for studies might appear restrictive to some readers. The decision to exclude studies based on language, publication date, and study types could potentially lead to the exclusion of relevant insights and perspectives, particularly if certain regions or disciplines are underrepresented in the chosen search platform.

Triangulation Phase: While the integration of insights from the EPIG event is a novel and interesting approach, more information could be provided about the nature of the event, the types of experts who participated, and how the event's outcomes were integrated into the review's analysis. This would help readers assess the validity and relevance of the event's contributions.

Discussion: The discussion section is thorough and provides important insights. However, the review might benefit from a more critical engagement with its own findings. Addressing potential limitations and biases in the scoping review process, as well as acknowledging any gaps in the literature or areas of uncertainty, would enhance the overall rigor of the paper.

Ethical Considerations: While the review addresses ethical considerations towards the end, it could delve deeper into potential ethical implications of pandemic response strategies, especially in terms of balancing public health needs, individual rights, and cultural sensitivities.

In conclusion, while the review provides valuable insights into COVID-19 pandemic management and governance, there are areas where additional clarity and critical engagement could enhance the overall quality and impact of the paper. Addressing these points could help make the research more accessible, relevant, and robust for readers in the field.

Author Response

Thank you for your valuable suggestions. All your suggestions were addressed.  

1. The methodology section has been reviewed. The selection of Scopus as database has been justified. The initial search on Scopus has been explained better and justified. 

2. The search string is now part of the main text, presented in Table 1. The decision to use Scopus was justified further in the text. The limitations section was expanded to take this into account. 

3. Discussions were expanded with limitations to take this into account. Please notice that Scopus is one of the largest transdisciplinary databases, therefore it is unlikely that some discipline may be underrepresented. In terms of other languages, only 5 papers were found in other languages, 4 in Spanish and 1 in Portuguese. Therefore, it is unlikely that we might have missed out some relevant information. Regarding publication dates, articles preceding 2019 were excluded, as COVID-19 started in 2019 and was officially recognised in 2020. Regarding the study types, only original research articles were included, to maintain a good quality of included evidence. Minor types such as letters to editors were excluded. This should not negatively affect the body of evidence we analysed. 

 4. To address your concerns, a paragraph has been added at pp4-5, l. 166-181). It gives more detail on speakers, content and how it was integrated into the discussion. This is also reflected by making the link more explicit in the discussion (throughout). 

 5.The discussions were extended and reformulated (see p.17 l.564-onwards to address this) 

 6. This was addressed in the Conclusion, please see pp. 19- 20. 

 7. Once again thank you for your valuable comments and feedback. We hope that by having addressed these points, the overall quality is now better. 

Round 2

Reviewer 1 Report

I think the current version is okay for publication. 

Author Response

Thank you for your valuable comments